# No Evidence Known Viruses Play a Role in the Pathogenesis of Onchocerciasis-Associated Epilepsy. An Explorative Metagenomic Case-Control Study

**DOI:** 10.3390/pathogens10070787

**Published:** 2021-06-22

**Authors:** Michael Roach, Adrian Cantu, Melissa Krizia Vieri, Matthew Cotten, Paul Kellam, My Phan, Lia van der Hoek, Michel Mandro, Floribert Tepage, Germain Mambandu, Gisele Musinya, Anne Laudisoit, Robert Colebunders, Robert Edwards, John L. Mokili

**Affiliations:** 1College of Science and Engineering, Flinders University, Adelaide, SA 5001, Australia; michael.roach@flinders.edu.au (M.R.); robert.edwards@flinders.edu.au (R.E.); 2Computational Sciences Research Center, Biology Department, San Diego State University, San Diego, CA 92182, USA; garbanyo@gmail.com; 3Global Health Institute, University of Antwerp, 2160 Antwerp, Belgium; Krizia.Vieri@uantwerpen.be (M.K.V.); robert.colebunders@uantwerpen.be (R.C.); 4Wellcome Trust Sanger Institute, Hinxton CB10 1RQ, UK; Matthew.Cotten@lshtm.ac.uk; 5MRC/UVRI and London School of Hygiene and Tropical Medicine, Entebbe, Uganda; p.kellam@imperial.ac.uk (P.K.); My.Phan@lshtm.ac.uk (M.P.); 6Centre for Virus Research, MRC-University of Glasgow, Glasgow G61 1QH, UK; 7Laboratory of Experimental Virology, Department of Medical Microbiology and Infection Prevention, Amsterdam UMC, University of Amsterdam, 1012 WX Amsterdam, The Netherlands; c.m.vanderhoek@amsterdamumc.nl; 8Provincial Health Division Ituri, Ministry of Health, Ituri, Congo; michelmandro8@gmail.com; 9Provincial Health Division Bas Uélé, Ministry of Health, Bas Uélé, Congo; floritepage@yahoo.fr; 10Provincial Health Division Tshopo, Ministry of Health, Tshopo, Congo; glmambadu@gmail.com; 11Médecins Sans Frontières, Bunia, Congo; mucinyagisele@gmail.com; 12Ecohealth Alliance, New York, NY 10018, USA; laudisoit@ecohealthalliance.org; 13Viral Information Institute, Biology Department, San Diego State University, San Diego, CA 92182, USA

**Keywords:** epilepsy, nodding syndrome, onchocerciasis, viruses, Democratic Republic of Congo

## Abstract

Despite the increasing epidemiological evidence that the *Onchocerca volvulus* parasite is strongly associated with epilepsy in children, hence the name onchocerciasis-associated epilepsy (OAE), the pathophysiological mechanism of OAE remains to be elucidated. In June 2014, children with unprovoked convulsive epilepsy and healthy controls were enrolled in a case control study in Titule, Bas-Uélé Province in the Democratic Republic of the Congo (DRC) to identify risk factors for epilepsy. Using a subset of samples collected from individuals enrolled in this study (16 persons with OAE and 9 controls) plasma, buffy coat, and cerebrospinal fluid (CSF) were subjected to random-primed next-generation sequencing. The resulting sequences were analyzed using sensitive computational methods to identify viral DNA and RNA sequences. *Anneloviridae, Flaviviridae, Hepadnaviridae* (*Hepatitis B virus*), *Herpesviridae*, *Papillomaviridae, Polyomaviridae* (Human polyomavirus), and *Virgaviridae* were identified in cases and in controls. Not unexpectedly, a variety of bacteriophages were also detected in all cases and controls. However, none of the identified viral sequences were found enriched in OAE cases, which was our criteria for agents that might play a role in the etiology or pathogenesis of OAE.

## 1. Introduction

A high prevalence of epilepsy has been reported in onchocerciasis-endemic regions with high ongoing or past *Onchocerca volvulus* transmission [1]. In these regions a majority of persons with epilepsy present with certain common characteristics such as an onset of epilepsy without any obvious cause—including no evidence of prior exposure to *Taenia solium*—in previously healthy children between the ages 3–18 years [1]. This type of epilepsy is now called “onchocerciasis-associated epilepsy” (OAE) and includes the Nodding syndrome and the Nakalanga syndrome [1]. While there is epidemiological evidence of an association between *O. volvulus* infestation and epilepsy [2], the physiopathology mechanism of OAE remains unknown.

Two cohort studies performed in an onchocerciasis-endemic region in the M’bam valley of Cameroon showed an association between the level of microfilarial load in children and the development of epilepsy [3,4]. This temporal and dose response relationship between *O. volvulus* infection and epilepsy suggests that the *O. volvulus* parasite could play a role in causing epilepsy. However, so far, neither *O. volvulus* microfilariae nor DNA from either *O. volvulus* or its endosymbiont *Wolbachia* have been detected in cerebrospinal fluid [5] or brain tissue of persons with OAE [6], which, if the parasite is playing a driving role in the disease, one might expect. Therefore, we explored alternative explanations for the link between onchocerciasis and epilepsy. One hypothesis is that female blackflies, the vector of onchocerciasis, may also transmit another pathogen [7]. Indeed, a high microfilarial load in a child also means the child has been frequently bitten by blackflies. A high number of bites increases the exposure risk to any other pathogen carried by the female blackfly, potentially including neurotropic viruses. Therefore, we used samples collected during a case control study in Titule, a locality in an onchocerciasis endemic area in the DRC, to carry out a viromic study on plasma, buffy coat, and cerebrospinal fluid to identify a potential viral cause for OAE [8].

## 2. Results

### 2.1. Characteristics of the Cases and Controls

A total of 18 cases and 9 controls were recruited in this study. However, using the OAE criteria [1] two persons were excluded from the study as cases. One was excluded because the first seizures started only at the age of 26. The other because the child was 3 years with his first seizures appearing at the age of 6 months (potentially febrile seizures). Therefore, 16 persons with OAE and 9 controls were included in the analysis. The median age and gender of cases and controls did not significantly differ (Table 1). Most cases presented only generalized tonic–clonic seizures (62.3%). In six (37.5%) absences were also reported, but in none of them were head nodding seizures observed. Half of the cases were mentally impaired, one (12.5%) with moderate mental impairment, and seven (43.75%) with important mental impairment. Three cases were considered to be stunted, one (6.3%) with moderate growth delay, and two (12.5%) with important growth delay, but none of them met the criteria of Nakalanga syndrome (born healthy but growth failure, wasting/emaciation, retardation of sexual development, and mental impairment [9]). Many cases (62.5%) complained of itching. Among cases, 14 (77.8%) were *O. volvulus* PCR skin snip positive while only one of five (20%) controls tested was skin snip positive.

### 2.2. Identification of Sequences from Known Viruses

The data analysis using Hecatomb identified 313,159 viral sequences over all samples (not including the two persons excluded from the study), of which 273,975 were considered high confidence hits (amino acid alignment support; e value < 1 × 10^−20^). For the OAE patients, all three sample types (plasma, buffy coat, and CSF) were sequenced. These three sources yielded very different quantities of viral sequences. Of the high-confidence hits, plasma samples accounted for 97.6% of all viral hit sequences, whereas buffy coat and CSF accounted for 1.7% and 0.7%, respectively. However, viral sequence hits in the buffy coat sample outnumbered the plasma sample for patient #11, and CSF outnumbered plasma for patient #16 (Appendix A).

Approximately 95.6% of the viral sequence hits were annotated as belonging to the families *Anelloviridae*, *Flaviviridae*, and *Hepadnaviridae* (Figure 1). All of the 79,362 sequence hits for *Hepadnaviridae* were identified as Hepatitis B virus and were mostly found in two patients (#4 and #11, Appendix A), and not in controls. Almost all *Flaviviridae* sequences (99.8%; 186,710 out of 187,013 sequence hits) were identified as Pegiviruses (Pegivirus, Pegivirus A, Pegivirus C/Hepatitis G Virus) and these were found in five of the controls and in patient #4. The 5697 *Anelloviridae* sequence hits were found across both case and control samples, in varying quantities, and most were either Torque teno virus (TTV) species (63.7%) or unknown *Anelloviridae* species (33.9%). Interestingly, 25 *Virgaviridae* sequence hits were observed in patient #6. These sequences matched Pepper mild mottle virus and Mosaic virus—both plant viruses. There were 1367 sequences that matched viral entries with no family taxonomic information; 1053 of these were annotated as various phage species and were spread across all patients.

## 3. Discussion

In this explorative study, we used a randomly-primed metagenomic approach to detect potential neurotropic viruses that could be associated with OAE. Upon removing low quality reads and reads from repetitive elements and host-contamination, only 1.7% of the remaining sequences showed homology to known virus sequences.

To ascertain the efficiency of the sequencing pipeline, the first step in data analysis was to examine the prevalence of anelloviruses, a ubiquitous virus in human viromes [10,11,12,13,14,15,16]. It was expected that the *Anelloviridae* would be detected and equally distributed among cases and controls. We previously detected TTV, an anellovirus, by PCR in over 50% of the general population in the DRC [17]. In the present study, of the three viral families that accounted for 95.6% of the high confidence sequence hits using Hecatomb analysis, the *Anelloviridae* family was the most prevalent, which was consistent with the previous observation [17]. In addition, TTV was identified in similar proportions among cases and controls.

There were no significant differences separating case and control samples that could be associated with OAE. None of the viruses that were identified, namely *Flaviviridae*, *Hepadnaviridae*, *Herpesviridae*, *Papillomaviridae*, *Polyomaviridae*, and *Virgaviridae*, have been known to be neurotropic, and could not be associated with the sequalae observed in children with OAE.

Although the data show no association with OAE, the presence of sequence reads of *Virgaviridae* from the buffy coat sample for patient #6 was intriguing (Figure 2, Appendix A). The *Virgaviridae* family are ssRNA viruses known to infect predominantly plants but are common in fecal samples of mammals, presumably from ingested plant material [18,19,20,21]. In human, *Virgaviridae* have previously been found to be the dominant viral type in the enteric virome of patients with coronary heart disease [22].

The *Papillomaviridae* species and Human polyomavirus 5 (*Polyomaviridae*) were also detected; however, the relatively low counts for sequences from these viruses and lack of statistical significance makes their biological relevance to OAE likely to be nonsignificant. The *Flaviviridae* were generally more prominent in the control samples and also could not be associated with the occurrence of OAE. There were two case patients with significant hits for Hepatitis B (*Hepadnaviridae*). These sequence hits were almost entirely found in plasma or buffy coat samples rather than CSF samples, and it is unclear how Hepatitis B might otherwise contribute to the onset of OAE.

A large number of sequences could not be classified using the methods and databases we have chosen, either due to poor or no matches of known database virus sequences, or from matching database sequences with an incomplete taxonomic status. Such sequences were previously called “dark matter” [23] and have been used as the blueprint for the discovery of novel viruses [23,24,25,26]. It is likely that in addition to known viruses that were identified in this study, there may still be other ones that could be characterized in silico and confirmed by independent methods such as PCR, culture, or electron microscopy.

Our study has major limitations. First, this was an explorative study with a very small sample size with only 9 controls for 16 cases. Cases and controls were from the same area but not matched for age and sex. It is also possible that if a neurotropic virus plays a role in the pathogenesis of OAE, this virus may only be present in the child prior to the development of the first seizures or shortly thereafter. Secondly, in all of our study participants, the onset of their first seizures occurred at least one year before the samples were obtained. If the neurologic manifestation appears after the clearance of the virus, it may be difficult to establish the etiology of OAE unless a well-designed longitudinal matched case-control study is conducted. Therefore, a much larger sample size of village, age, and sex matched controls will be needed, and only children with very recent onset of epilepsy should be included. Moreover, next to viromic studies of children with OAE, a viromic study of the salivary glands of female blackflies and of the microbiome of the *O. volvulus* parasite should be considered. At the same time, further research is needed to explore whether the *O. volvulus* parasite through an indirect mechanism, e.g., through the excretion/secretion of certain proteins or through the induction of an auto-immune process, is able to induce epilepsy. In addition, future research should also focus on potential co-etiological factors, including genetic ones. A small study using samples of patients with nodding syndrome and controls from South Sudan identified certain HLA haplotypes that were protective and others that were associated with susceptibility [27]. A larger HLA study will need to confirm these findings, but also a whole exome sequencing study should be considered. Moreover, in case control studies, the use of a high-resolution SNP array could exclude genetic causes of epilepsy.

In conclusion, our pilot metagenomics study was unable to provide evidence that known viruses play a role in the pathogenesis of OAE. However larger case control studies that include CSF from cases as well as control individuals are needed to investigate whether known viruses such as those identified in this study, or other unknown viruses, play a pathogenic role in OAE.

## 4. Materials and Methods

### 4.1. Study Design and Procedures

This study was a sub-study of a case-control study performed in June 2014, in the city of Titule, in Bas-Uélé Province of the DRC to determine risk factors for epilepsy [8]. A door-to-door epilepsy prevalence study, performed in Titule concomitantly with the case-control study, documented a 2.3% epilepsy prevalence [28]. The case-control study enrolled 59 cases and 61 controls [8]. The inclusion criteria for cases included a history of at least 2 episodes of unprovoked generalized tonic seizures and absence of known etiology. Nine healthy individuals with no clinical symptoms, who lived in the same or nearby villages and who did not belong to a family with cases of epilepsy, were recruited as controls. A written informed consent was obtained from each participant in his/her native language by physicians (MM, GMa, GMu) using a standardized mixed-methods questionnaire. After the interview, cases and controls were examined by a physician (MM, GMa, GMu, or RC). In persons with a history of seizures, questions were asked about the year of onset of the seizures, whether the person was healthy before the development of the seizures, the type of seizures, and the frequency of seizures. Moreover, their mental impairment was assessed by asking the parent/care giver the following three questions:Does the person have major difficulties expressing themselves?Does the person get lost in the village?Does the person understand questions?

If a response to one of these questions was “yes”, we considered the person to be mentally severely impaired. If the person had only minor difficulties to express themselves the person was considered to be moderately mentally impaired. On physical examination, cases and controls were assessed for skin abnormalities that are characteristic of onchocerciasis. Height and weight were measured using a stadiometer and a digital scale, and these measurements were used to calculate the body mass index (BMI, kg/m^2^). In persons with epilepsy, a lumbar puncture was performed by a physician (GMu) who had received special training in neurology. After the procedure, patients were able to rest and received paracetamol.

### 4.2. Testing for Onchocerciasis

Blood samples were collected from all cases and controls in heparinized collection vials and were tested for OV IgG4 antibodies with a rapid diagnostic test (Ov16 Standard Diagnostics, Inc, Alere SD BIOLINE, Gyeonggi-do, Korea). A skin snip was taken from the left and right iliac crests of cases and controls with a Holtz corneoscleral punch (2 mm) and stored in 90% ethanol to be tested for OV by an in-house PCR method.

### 4.3. Metagenomic Analysis

Samples from cases (plasma, buffy coat, and cerebrospinal fluid (CSF)) and controls (plasma) were subjected to nucleic acid extraction to detect viral sequences and then to random-primed next-generation sequencing as previously described [29] using a method to deplete un-encapsidated host and bacterial DNA and amplify viral nucleic acid. Briefly, 110 µL of cerebrospinal fluid, plasma, or buffy coat collected from cases and plasma from controls were spun down to remove cells, and 100 µL of the supernatant was subjected to DNase treatment to eliminate background cellular DNA with 20 U TURBO™ DNase (Ambion, Life Technologies, Carlsbad, CA, USA). Nucleic acids were extracted from the pre-treated samples as described by Boom [30]. In order to subsequently detect RNA viruses a reverse transcription with 200 U of Superscript II (Invitrogen, Life Technologies, Carlsbad, CA, USA) and non-ribosomal hexamers [31] was performed followed by a second strand synthesis with 5 U of Klenow fragment (3′–5′ exo-) (New England Biolabs, Ipswich, MA, USA) and 7.5 U of RNase H (New England Biolabs, Ipswich, MA, USA). Samples were purified by a phenol chloroform extraction and ethanol precipitation. Subsequent Illumina MiSeq library prep on the dsDNA was performed as described [32].

A total of 63 libraries (2 × 250 bp) were prepared and sequenced. Viral sequences for each sample were identified from the sequencing data using the Hecatomb pipeline (github.com/shandley/hecatomb; accessed 3 February 2021). Briefly, Hecatomb removes spurious sequences (such as adapters and primers) from viral metagenomes, filters the sequences to remove human and bacterial sequences, and then clusters the remaining sequences. Taxonomic information is assigned to the remaining viral sequences using MMseqs2 [33]. The following set of reference nucleotide and amino acid viral databases are used: SINEBase [34] and a manually compiled collection of primer and adapter sequences for non-host contaminant removal; the Human reference genome GRCh38 [35] for host contaminant removal; UniProtKB viral protein sequences and the UniRef50 database [36]; and RefSeq Viral nucleotide database [37] masked using the UniVec database [38]. MMseqs2 produces a similar alignment output to BLAST, so the expected value, alignment length, and alignment percentage identities for all hits were retained together with the taxonomic assignments and database types. Sequences identified by Hecatomb were filtered to ensure only high-confidence hits were used in the analysis. This involved removing sequences lacking amino-acid alignment support to known viral proteins and applying an e value cutoff of 1 × 10^−20^ to remove low quality alignments. The data were analyzed and visualized in R with ggplot2.

## Figures and Tables

**Figure 1 pathogens-10-00787-f001:**
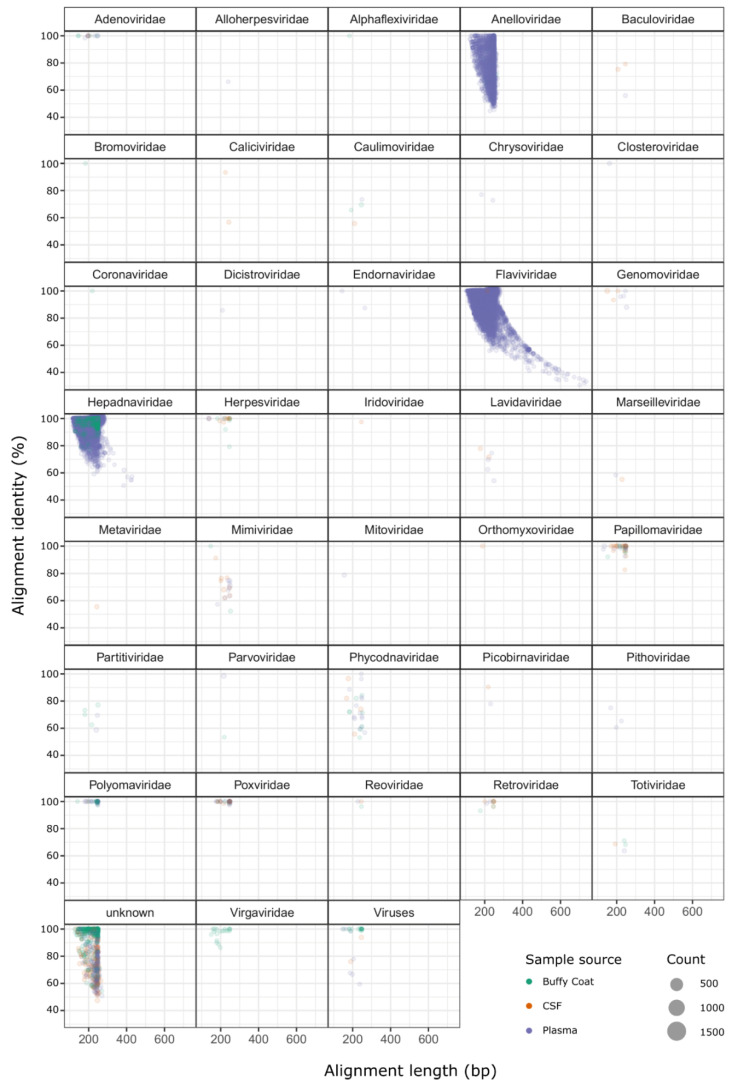
Alignment length and percentage identity to reference source for all high-confidence viral hit sequences. Separate plots are shown for each viral family identified, points are scaled according Appendix A. The hit sequences for each case and control patient were plotted for the above virus families, separated by family and sample source, and Student’s t-tests were performed (comparing the case and control plasma sample counts) to determine a statistical significance for any differences between case and control groups (Figure 2). Hit sequence counts were low for *Herpesviridae, Papillomaviridae*, Human polyomavirus, and *Virgaviridae* (ranging between 25 and 93 hits across all samples), and no statistically significant differences were observed. The differences between case and control groups for *Anelloviridae* and Hepatitis B were also not significant. *Flaviviridae* was the only family with a difference approaching statistical significance (*p*-value = 0.078), with mean hit sequence counts of 616 for case patients and 22,015 for control subjects.

**Figure 2 pathogens-10-00787-f002:**
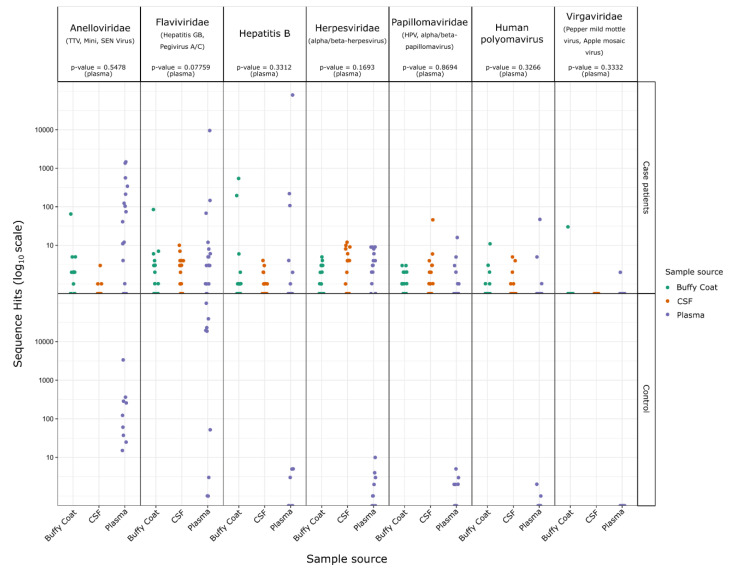
Sequence hit count differences for case and control patient samples. Hit counts are show for case and control patients for the indicated viral families and separated by sample source. *p*-values are indicated for T-tests of the case versus control for each virus family using the counts from plasma samples only (as Buffy coat, and CSF samples were not collected for control patients).

**Table 1 pathogens-10-00787-t001:** Characteristics of cases and controls.

Clinical Characteristics	Cases (*n* = 16)	Controls (*n* = 9)	*p*-Value
Age, median (IQR)	14.5 (11.7–19.7)	16.0 (10.5–24.5)	0.820
Male *n* (%)	7 (43.7)	3 (42.9)	0.778
Body weight (kg) mean (S.D.)	39.0 (14.0)	40.7 (14.9) *	1.000
Height (cm), mean (S.D.)	145.0 (19.0)	150.5 (16.0) *	0.673
Body mass index, mean (S.D.)	17.7 (3.5)	17.3 (2.7) *	0.923
Age of epilepsy onset, median (IQR)	10.0 (9.0–13.0)	NA	NA
Only generalized tonic–clonic seizures n (%)	10 (62.3)	NA	NA
Generalized tonic–clonic seizures and absences *n* (%)	5 (31.2)	NA	NA
Only absences *n* (%)	1 (6.2)	NA	NA
Mental disorder *n* (%)	8 (50)	NA	NA
Stunting *n* (%)	3 (18.7)	NA	NA
Onchocerciasis skin lesion (“leopard skin legions” both legs) *n* (%)	3 (18.75)	NA	NA
Itching *n* (%)	10 (62.5)	1 (11.1) *	0.061
Onchocerciasis nodules	2 (12.5)		
Skin snip *O. volvulus* PCR positive n (%)	14 (77.8)	1/5 (20)	0.009
*O. volvulus* IgG4, *n* (%)	10/15 (66.6)	NA	NA

IQR = Interquartile range, SD = Standard deviation. * Data available only in 7 controls.

## Data Availability

Sequencing data were deposited in the Sequence Read Archive (National Center for Biotechnology Information) under the BioProject accession PRJEB9580. The summary table used for figures and analyses is available at doi.org/10.5281/zenodo.4571692 (https://zenodo.org/record/4571692, accessed on 2 March 2021).

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
