# Peer review of "No Evidence Known Viruses Play a Role in the Pathogenesis of Onchocerciasis-Associated Epilepsy. An Explorative Metagenomic Case-Control Study"

_pathogens, 2021, doi:10.3390/pathogens10070787_

Round 1
Reviewer 1 Report
Roach et al. aim to investigate if viruses might play a role in OAE. Therefore, plasma, buffy coat and CSF were obtained from volunteers in DRC to identify virus spp. using NGS. The results are interesting and the manuscript is well written. However, some issues remain unclear and need to be clarified before publication.
Major comments:
1) Indeed, the pathophysiological mechanisms of OAE are unknown and thus this study is very important to elucidate the role of viruses. However, the authors should discuss and also highlight that it is suggested that autoimmune responses might play a crucial role for OAE. Indeed, several studies show that filarial-driven immune responses and regulation of host immunity influences concomitant infections. Indeed, a recently published article (Chetty et al. 2021, Cell Host and Microbes) revealed that systemic filarial infection can influence virus-driven pathology. Thus, did the authors consider that coinfections rather than pure presence of viruses might play a role, especially in regards to unbalanced immune responses accompanied with genetic background.
2) Since the authors took skin snips information about microfilarial numbers would be important.
3) The authors stated that most of the viral hits were obtained in plasma samples, and only 0.7% accounted for CSF. However, the results from CSF are, in my opinion, very important since this might reveal a direct connection to OAE. Indeed, several virus spp. are present in the CSF of case patients but not in control (Figure 2 and Figure S2). Are there any significant differences? In the manuscript it is stated that differences were only assessed of the plasma samples. In general, it seems that in buffy coat and CSF most virus spp. are present in case patients but absent in the controls. I think this is a very important finding.
4) In regards to the previous point, why did the authors not obtain household/family members as controls.
5) Figure S1 the controls are missing. What is the star meaning in patient 4.
Minor comment:
1) Figure S2: There is a cross above the control patients
2) line 201-202: What is MM, GMa, GMu and RC
3) In my opinion, it would be nice for the reader to separate the method section according to the applied methods (such as patient classification, Ov diagnosis, PCR etc)
4) line 212-213: with skin abnormalities do the authors mean nodules? Most of infected individuals are asymptomatic and do not develop skin diseases
Reviewer 2 Report
This is a well-written manuscript addressing the search for a potential viral origin of, or involvement in the so-called Onchocerciasis-associated epilepsy.
The study was conducted by analysis of samples from patients (plasma, CSF, buffy coat) and controls (plasma) collected during a previous study in an endemic onchocerciasis region in Congo carried out in 2014 and reported in 2016.
General remarks
There were no obvious differences between cases and controls with respect to the results of plasma analysis. Although the study population denominators were small (16 cases, 9 controls), if the hypothesis was that a single viral cause of mechanism is responsible for most of the cases, one could conclude that the current results do not point to one among the studied ones.
However, the authors are right by suggesting a future prospective cohort study in view of the successive time differences between supposed causal infection/infestation, onset of epilepsy, and time point of the study in 2014. For example, the difference between average age at onset of epilepsy (10) and at sampling (14.5) was 4.5 years, whereas the age at infection/infestation is not given, possibly not assessed/assessable. From this point of view, it was unlikely a priori that the current study would have identified a single viral cause.
Percentages in tables and text are given with a precision of one decimal, e.g. 62.3%, suggesting it is not 62.2% or 62.4%. Since all denominators are much smaller than 100 (16 and 9 respectively, or less), zero decimal is correct and precise (in this example 62%), for denominators between 100 and 1,000 one decimal, for denominators between 1,000 and 10,000 two decimals, and so on.
In the introduction, the authors added an interesting argument, that the chances of infection with other potential pathogens than Onchocerca volvulus is related to the number of bites by the vector Black fly.
This raises implicitly the question as to whether there was or not a relation between the percentage of Onchocerca volvulus positive skin snip tests and viral sequence test results among cases in CSF and buffy coat, and among all study participants (cases and controls) in plasma.
In the discussion they may also consider the possibility that another pathogen than Onchocerca volvulus, if any, may cause both epilepsy as well as increased susceptibility to onchocerciasis, in parellel and not by cause and effect.
Finally, if the authors intend to start a prospective cohort study, I strongly suggest to include whole exome sequencing (WES) and high resolution SNP array for genetic causes of epilepsy, for a number of reasons: 1) genetic causes of epilepsy have never been studied in this kind patient populations, whereas in general genetic analysis of complicated epilepsies with significant comorbidities provide a very high yield up to more than 50% of cases, 2) exclusion of cases due to a genetic cause leads to greater power to detect remaining environmental causes including infections and infestation, and 3) genetic analysis may help tot identify genetic susceptibilities to environmental causes.
Grammar and text
Page 3, line 91: “… were and sequenced …”: delete “and”?
Round 2
Reviewer 1 Report
I thank the authors for the clarification and the response of my questions and issues. However, in regards to the sample types, except that most of the viral hit sequences were found in the plasma in comparison to CSF and buffy coat, the authors cannot draw any conclusion about the CSF and buffy coat results since controls are missing. In my opinion the CSF and buffy coat results will not give any additional information and thus should be deleted or at least placed into the supplements. Moreover, the authors should focus on plasma and amend the abstract/conclusion since in my opinion the manuscript suggests that also no viral sequence were found in CSF samples that could be connected to OAE, but this is not possible to conclude from these data sets since the controls are missing.
Round 3
Reviewer 1 Report
I thank the authors for the additional explanation. I think the manuscript is now suitable for publication in Pathogens.